# Patterns and Characteristics of SKYLINE-Lumipoint Feature in the Catheter Ablation of Atypical Atrial Flutter: Insight from a Novel Lumipoint Module of Rhythmia Mapping System

**DOI:** 10.3390/jpm12071102

**Published:** 2022-07-04

**Authors:** Cheng-Hung Li, Li-Wei Lo, Ankit Jain, Yu-Cheng Hsieh, Yenn-Jiang Lin, Shih-Lin Chang, Fa-Po Chung, Yu-Feng Hu, Tze-Fan Chao, Jo-Nan Liao, Ting-Yung Chang, Chin-Yu Lin, Isaiah Carlos Lugtu, An Nu-Khanh Ton, Shin-Huei Liu, Wen-Han Cheng, Chih-Min Liu, Cheng-I Wu, Shih-Ann Chen

**Affiliations:** 1Division of Electrophysiology, Cardiovascular Center, Taichung Veterans General Hospital, Taichung 407, Taiwan; lichcil@yahoo.com.tw (C.-H.L.); ychsieh@vghtc.gov.tw (Y.-C.H.); 2Institute of Clinical Medicine, Cardiovascular Research Center, National Yang Ming Chiao Tung University, Taipei 112, Taiwan; linyennjiang@gmail.com (Y.-J.L.); ep.slchang@msa.hinet.net (S.-L.C.); marxtaiji@gmail.com (F.-P.C.); huhuhu0609@gmail.com (Y.-F.H.); eyckeyck@gmail.com (T.-F.C.); care1980@gmail.com (J.-N.L.); clouaa@gmail.com (C.-Y.L.); 3Department of Data Science and Big Data Analytics, Department of Financial Engineering, Providence University, Taichung 433, Taiwan; 4Division of Cardiology, Department of Medicine, Taipei Veterans General Hospital, Taipei 112, Taiwan; dr.ankitjain@gmail.com (A.J.); tingyungchang@gmail.com (T.-Y.C.); isaiahlugtu@gmail.com (I.C.L.); hailam2310@gmail.com (A.N.-K.T.); shliu9@vghtpe.gov.tw (S.-H.L.); hill55772003@gmail.com (W.-H.C.); sasuke9301108@hotmail.com (C.-M.L.); shawnwu64@gmail.com (C.-I.W.); 5Department of Post-Baccalaureate Medicine, College of Medicine, National Chung-Hsing University, Taichung 402, Taiwan

**Keywords:** atypical atrial flutter, catheter ablation, high-density mapping system, global activation histogram, Lumipoint algorithm

## Abstract

Background: Atypical atrial flutter (aAFL) is not uncommon, especially after a prior cardiac surgery or extensive ablation in atrial fibrillation (AF). Aims: To revisit aAFL, we used a novel Lumipoint algorithm in the Rhythmia mapping system to evaluate tachycardia circuit by the patterns of global activation histogram (GAH, SKYLINE) in assisting aAFL ablation. Methods: Fifteen patients presenting with 20 different incessant aAFL, including two naïve, six with a prior AF ablation, and seven with prior cardiac surgery were studied. Results: Reentry aAFL in SKYLINE typically was a multi-deflected peak with 1.5 GAH-valleys. Valleys were sharp and narrow-based. Most reentry aAFL (18/20, 90%) lacked a plateau and displayed a steep GAH-valley with 2 GAH-valleys per tachycardia. Each GAH-valley highlighted 1.9 areas in the map. Successful sites of ablation all matched one of the highlighted areas based on GAH-valleys < 0.4. These sites corresponded with the areas highlighted by GAH-score < 0.4 in reentry aAFL, and by GAH-score < 0.2 in localized-reentry aAFL. Conclusions: The present study showed benefits of the Lumipoint^TM^ module applied to the RhythmiaTM mapping system. The results were the efficient detection of the slow conduction, better identification of ablation sites, and fast termination of the aAFL with favorable outcomes.

## 1. Introduction

Atypical atrial flutters (aAFL), frequently observed due to both substrate and ablation scars, can be an undesirable sequelae after the catheter ablation of atrial fibrillation (AF) or may develop with structural heart disease (SHD) [1,2]. Apart of prior ablation procedures, aAFL may turn into a primary arrhythmia following cardiac surgery (scar-related), congenital defect or other underlying atrial diseases [3,4]. Such events of aAFL, or atrial tachycardia (AT), either post–AF ablation or scar-related, are difficult to manage medically and frequently recur after electrical cardioversion [5,6]. On the other hand, aAFL may represent a more organized atrial substrate on the continuum between AF and sinus rhythm [7,8].

Ablation of aAFL is challenging with limited success despite lengthy procedures [6,9,10]. In aAFL ablation, high-density mapping (HDM) system reveals complex activation patterns and extends the common knowledge of AFL. It provides more information about tissue and structure than simple categorization of them as focal, localized reentry or macro-reentry [11,12]. The standard Rhythmia^TM^ (Boston Scientific, Marlborough, MA, USA) mapping system consisted of a 64-pole basket mapping catheter (IntellaMap Orion; Boston Scientific) that could yield maps with a previously unattained resolution [13,14,15]. Lumipoint^TM^ is a novel set of software incorporated in the Rhythmia^TM^ mapping system to detect all activations present in every electrogram (EGM) irrespective of local activation time [16]. Very few reports are available on the application of Lumipoint^TM^ in atrial procedures [17,18] not to speak of complex aAFLs. Here, we aimed to evaluate the feasibility and utility of the Lumipoint^TM^ algorithm on the ultra-rapid high-density Rhythmia^TM^ mapping system and novel to map, characterize, and guide catheter ablation of aAFL with complex electrophysiology in patients with SHD and complicated underlying substrates.

## 2. Materials and Methods

### 2.1. Study Population

There were 38 atrial flutter (AFL) patients undergoing mapping and ablation guided by HDM (high-density mapping) with Boston Rhythmia^TM^ system treated at two tertiary medical centers: Taichung Veterans General Hospital (VGH) and Taipei VGH between May 2017 and October 2019. Of them, 15 patients presented with symptomatic, drug refractory incessant aAFL during the procedure were included in the study. Other patients with only CTI (cavotricuspid isthmus)-dependent AFL were excluded. This study was approved by the institutional review committees of the two respective hospitals (IRB ID: CE20174B, Taichung VGH and 2020-06-009BC, Taipei VGH). All patients gave written informed consent beforehand.

### 2.2. Electrophysiological Study and HDM with Rhythmia System

For all patients, anti-arrhythmic drugs, except amiodarone, were discontinued for >5 half-lives before ablation. They all received adequate anticoagulation treatment for >3 weeks before ablation and anticoagulants withheld on the day of procedure. Trans-esophageal echocardiography was routinely performed before the procedure to exclude left atrial thrombus. The ablation procedure was performed during their fasting state after the written informed consent had been obtained. An activated clotting time > 300 s was maintained throughout the procedure. The general catheter settings consisted of the following: (a) a fixed curve 5F (Viking™, 5 mm spacing; Boston Scientific Corporation, Marlborough, MA, USA) or a steerable 6F decapolar diagnostic catheter (Inquiry™, 5 mm spacing; St. Jude Medical, St. Paul, MN, USA) positioned in the coronary sinus, serving also as the reference of the Rhythmia™ 3-D electroanatomical mapping system (Boston Scientific Corporation, Marlborough, MA, USA); (b) an expandable, open irrigated 64-pole basket mapping catheter (IntellaMap Orion™, Boston Scientific) comprising of 8 splines with 8 electrodes (electrode spacing 2.5 mm, electrode surface area 0.4 mm^2^); and (c) an open-irrigated tip, bidirectional mapping and ablation catheter (IntellaNav Open-Irrigated, Boston Scientific).

HDM was performed with Rhythmia™ mapping system, which allowed real-time automated signal analysis. Automated mapping was done through a continuous mapping mode via user-defined beat acceptance criteria based on the following: (a) cycle length (CL) stability, (b) activation time difference variations between the coronary sinus EGMs, (c) mapping catheter motion, (d) EGM stability, (e) catheter tracking quality, and (f) respiration gating [18]. Details of the EGM annotations have been reported earlier [15].

### 2.3. Definition and Classification of aAFL

The condition of aAFL was evaluated based on the HDM data. According to timings of the bipolar activation map, a propagation map was generated. Wave-front propagation was visualized by advancing a 10-ms window of activation along the time axis. Based on the propagation map alone, the number and type of atrial reentrant circuits accounting for 90% of the tachycardia cycle length (TCL) were determined [19]. Three types of aAFL were classified: macro-reentry, localized reentry (micro-reentry) and multiple-loop. In this study, we defined a localized reentry as an AFL reentry circuit confined within a limited region or to one atrial wall. A macro–reentry was those with atrial EGMs spanning the whole CL of the tachycardia. Except CTI-dependent AFL, macro-reentry AFL included all typical anatomical AFL, like peri-mitral, or roof-dependent AFL. A multiple-loop AFL was defined as those composed of >1 independent activation circuits sharing a common conduction isthmus, including dual-loop AFL and ‘figure-of-eight’ AFL. We did not use the traditional methods (like entrainment mapping) to determine the site of activation in cases of unstable AFL, AFLs with short CL (<240 ms), and AFL in a repeated AF ablation.

### 2.4. Catheter Ablation

Selection of the appropriate ablation target was solely based on reentry circuits and isthmus visualization, without the entrainment mapping. In most aAFL cases, this target is the narrowest bridge of conducting tissue between scars or anatomical obstacles. Locations selected for ablation are defined as the practical ablation sites. Sites with aAFL termination during ablation were tagged as successful. Radiofrequency current was applied with a maximum power of 30 to 35 W and an irrigation rate of 17 to 30 mL/min for up to 120 s not exceeding an upper temperature limit of 42 °C. Procedural success of radiofrequency application was defined as termination of the tachycardia and the inability to re-induce the tachycardia by rapid atrial pacing and programmed atrial stimulation. The end point of this study is acute termination with non-inducible aAFL after catheter ablation.

### 2.5. Offline HDM Lumipoint Algorithm and SKYLINE Analyses

Maps were analyzed offline after data acquisition using the Rhythmia™ software. The Lumipoint™ algorithm was applied. These included the feature of ‘activation search’ (highlighting regions of the map containing electrogram activity in a given time-of-interest period), the feature of ‘complex activation’ (highlighting regions of the map that were activated within the time-of-interest period and were also showing multiple components of activation), the feature of ‘SKYLINE graph’ (reflecting the size of the activated region throughout the mapped window). Lumipoint™ confidence was applied to determine the likelihood or probability of an EGM having a genuine biological activation at a given time [20].

SKYLINE, one of the Lumipoint^TM^ features, is a display option on full chamber activation, converting 3D spatial and temporal activation data into an intuitive 2D tracing. The information is displayed through the global activation histogram (GAH) showing, in our cases, the atrial surface activation pattern throughout aAFL CL. It plotted the relative proportions of the atrial surface region activated at each time point throughout the entire CL. The histogram was in a normalized display scale (defined as the GAH-score), ranging from 0 to 1.0. Peaks appeared whenever a large part of the atrium had been activated, whereas valleys corresponded to smaller parts of activation [18].

Theoretically, a decrease in the activation area, which is described as a valley in the global activation histogram (GAH-Valley) could indicate an area of slow conduction, like the isthmus of the tachycardia. This area reflected entry of the activation wave-front, before it narrower, and exited. Following the reported method [18], we also set a 30-ms unit time interval and examined the relation between the GAH-Valleys, activation patterns and electrophysiological properties on the activation map. Several activation features were identified based on the high-resolution activation map: normal conduction, slow conduction, wave-front collision, and lines of block. Finally, the practical ablation sites tagged during the procedure were compared to the highlighted areas corresponding to GAH-Valleys. Figure 1 demonstrated the stepwise approach to apply Lumipoint module and Skyline analysis in aAFL ablation.

### 2.6. Statistical Analyses

Continuous variables were represented as mean ± standard deviation (SD). Median values and 25th and 75th percentiles were given in case of non-normal distributions. Categorical variables were represented as number (percentage). Chi-square or Fisher exact test was used to compare categorical variables. A two-sided *p* < 0.05 was considered significant.

## 3. Results

### 3.1. Patient Clinical Characteristics and Outcomes

A total of 20 aAFLs in 15 patients were successfully mapped and treated with ablation. Clinical characteristics of these patients are listed in Table 1, and their aAFL characteristics are also shown in Table 1. Twelve (80%) patients had undergone more than one ablation before. Except 2 naive aAFLs, 13 patients with prior procedures included cardiovascular surgery in 7 cases (47%) and previous AF/AFL ablations in 6 cases (40%). One case (7%) had an aAFL after the prior cardiovascular surgery and then later received another LA ablation. Among these 15 patients, 20 spontaneous aAFLs were mapped and diagnosed into one of the three types: macro-reentry (*n* = 14), localized reentry (*n* = 5), and multiple-loop reentry (*n* = 1). All aAFLs were successfully terminated at the practical ablation sites during radiofrequency application. At a mean follow-up period of 12.5 ± 9.3 months, 5 of 15 patients (33%) developed AF/AFL recurrences (4 aAFLs,1 AF).

### 3.2. GAH-Valleys and Corresponding Highlighted Areas

Table 2 shows numbers of lowest GAH-Valley (GAH < 0.2) per AFL: 2 (1–2) in macro-reentry aAFL, 1 (1–1) in localized reentry aAFL, and 1 in multiple-loop reentry aAFL. The numbers of highlighted areas in lowest GAH-Valleys were 51 for macro-reentry, 11 for localized reentry, and 1 for multiple-loop aAFL.

### 3.3. Activation Wave-Front Characteristics of Highlighted Areas Corresponding to GAH-Valleys

In 14 aAFLs of macro-reentry, 51 areas were highlighted based on 27 lowest GAH-Valleys (Table 2). The electrophysiological properties of the lowest GAH-Valley corresponding area was shown in Table 3. Twenty-three areas showed signs of slow conduction, 20 of wave-front collision, 5 of lines of block, and 3 of others, i.e., wave-front breakout. Of the 23 slow conductions, 19 (83%) occurred within the aAFL circuit (Table 3A).In 5 aAFLs of localized reentry (Figure 2), 11 areas were highlighted based on 6 lowest GAH-Valleys; 6 of them showed slow conductions, and the remaining 5 were wave-front collisions. Of the slow conductions, all (6, 100%) occurred within the aAFL circuit (Table 3B).In the single aAFL of multiple-loop reentry (Figure 3), one area that was highlighted based on the lowest GAH-Valley demonstrated a slow conduction. The slow conduction lay on the entrance, rather than the critical isthmus, of the aAFL circuit (Table 3C).

### 3.4. Practical Ablation Sites and Outcome

In 12 of 20 reentry aAFLs (12/20, 60%), final successful sites matched to the highlighted areas corresponding to lowest GAH-Valleys (Figure 4A & Table 2). In these 20 aAFLs, the successful ablation sites consistently located at areas with GAH-Valley scores < 0.4: 60% of all 20 successful sites in GAH-Valley < 0.2 and 40% in GAH-Valley 0.2–0.4 (Figure 4B). In localized reentries, successful ablation sites corresponded to GAH-Valley < 0.2 in 4/5 aAFLs (80%). In contrast, successful ablation sites corresponded modestly to GAH-Valley scores < 0.2 (8/14, 57%) for macro-reentry aAFLs. Areas highlighted with GAH-Valley scores < 0.2 (vs. GAH 0.2–0.4) showed a trend toward better predict of successful ablation sites in the localized reentries, than with macro- and multiple-loop reentries (*p* = 0.29) (Figure 4B).

Right panel shows the reentry circuit in the LA anterior wall between the left atrial appendage (LAA) and aortic groove. Left panel shows that in SKYLINE, a GAH-Valley with a score of <0.2 was identified. Middle panel shows the very beginning of the valley highlighted two areas: a critical slow conduction between LAA and aortic groove (solid bold line with arrow), and a line of block in LSPV (dashed line with round head). Ablation on the slow conduction terminated the aAFL.

Left panel shows the SKYLINE pattern with 2 GAH-Valleys. A: The deepest valley with a GAH-score < 0.2 (red hollow arrow) highlighting one area (site 1), representing slow conduction but not at the critical isthmus. Ablation at site 1 did not terminate the tachycardia. B: Instead, a second valley with GAH-score 0.2 to 0.4 (red hollow arrow) highlight three areas (sites 1 to 3). Site 1 represents the slow conduction that is the predominant part of the macro-reentry circuit. Site 2 represents a wave-front collision of two different activations. Site 3 again represents a slow conduction, but it is merely a bystander to the macro-reentry circuit. Ablation at site 1 successfully terminated the tachycardia.

## 4. Discussion

### 4.1. Major Findings

This is the first study reporting experiences using a HDM with the novel LUMIPOINT module for mapping and ablation of the complex aAFL in patients with post-AF ablation or prior surgery in SHD. We found that:SKYLINE feature of most reentry aAFLs displayed a multi-deflected peak with an average of 1.5 GAH-valleys.In the aAFL patients, these GAH-valleys were usually steep, instead of solitary peak and plateau. Its corresponding highlighted area harbored the comparable incidence of slow conductions (30, 47%) and wave-front collisions (26, 41%).In localized reentry related aAFLs, areas with a GAH-valley <0.2 in SKYLINE usually indicated the successful ablation site.In macro-reentry, successful sites were areas with a GAH-valley score <0.4, not limited to areas with a GAH-valley <0.2.Taken together, in these 20 aAFLs, successful ablation sites all occurred at areas showing GAH-valleys <0.4.

Such features were helpful in identifying potential ablation targets.

### 4.2. Patterns and Characteristics of SKYLINE

We found that the SKYLINE patterns did reflect the characteristics of the underlying anatomical substrates. The more deflections found in SKYLINE, the more complex the aAFL circuits appeared. This suggests that the complexity of underlying anatomy or substrates caused complex aAFL circuits.

The LumipointTM algorithm, based on anatomical and timing characteristics of the circuit, identifies any region where the EGM timing falls within a narrow area. This information on full chamber activation displays spatial and temporal activation patterns through the GAH, which shows the relative proportion of the atrial surface area that is activated at each time point throughout the entire CL [16]. In a simple reentry aAFL circuit, where the activation wave-front propagates throughout the entire chamber in one direction without wave-break from anatomical or substrate scar, SKYLINE pattern is a solitary peak with a wide-based GAH-Valley (Figure 1). And the corresponding GAH-Valley scores are typically <0.2, a finding suggesting the critical isthmus of conduction. On the contrary, in a complex reentry aAFL circuit in which activation wave-front propagates and diverges into several wave-fronts like with multiple iatrogenic scar or an anatomic barrier, SKYLINE patterns would be complex activation peaks with multiple deflections and steeper GAH-Valleys, as shown in the Figure 1, Figure 2, Figure 3 and Figure 4. 

### 4.3. Valley in SKYLINE (GAH-Valley)—Conduction ‘Deceleration’

Valley in the SKYLINE (GAH-Valley) reflects a smaller surface of activation in an aAFL and suggests an area of either slow conduction, lines of block, or wave-front collision. We called this a steep drop in activated areas (GAH-Valley) a conduction ‘‘deceleration’’. Hence, multiple iatrogenic scars (like post-AF ablation or surgical scar in SHD) can add on top of the original substrate, forming a complex anatomy/substrate, leading to the conduction abnormality of ‘‘deceleration’’. Multiple time points within one tachycardia cycle in an aAFL may therefore be identified as ‘deceleration’, reflecting multiple regions in the atrium highlighted by GAH-Valley. At a specific time point within the cycle, GAH values increase along with the activation wave-front breaking itself into several directions. As the dominant wave-front propagates and other bystander wave-fronts stop the activation due to wave-front collision or line of block, GAH scores can drop after these bystander activation wave-fronts. This explains why critical isthmus is not always in the deepest GAH-Valley. Instead, it is in the relatively deeper GAH-Valley. As the activation wave-front diverges and breaks due to underlying scar or anatomical barriers, the critical isthmus may coincide with other bystander wave-fronts in certain time point within the cycle. Consequently, the critical isthmus may be located in a relatively deep GAH-Valley with scores > 0.2, and not necessarily in the deepest GAH-Valley.

The SKYLINE pattern of a reentry aAFL in a complex substrate or anatomy usually displays many deflections of the GAH peaks, reflecting the complex wave-front properties within the aAFL circuit. This explains why a complex aAFL typically is associated with multiple deflections in the SKYLINE pattern with very sharp or steep GAH-valleys. As shown in our study (Table 3), more highlighted areas (drop in activated areas) were found in a macro-reentry aAFL compared with a localized reentry. Also, we found more areas of slow conduction in the circuit and more wave-front collisions outside the circuit in aAFLs of macro-reentry. Therefore, conduction abnormalities located within and outside the circuit tended to exist in an aAFL because of the complex anatomy or substrates. These more frequent presence of conduction abnormalities, or ‘‘decelerations’’, created more deflections in the SKYLINE patterns.

As reported by Takigawa et al. [18], the SKYLINE pattern of focal AT systemically displayed a plateau with GAH-valley scores < 0.1 covering >30% of the CL. They also found that most reentrant ATs (98.9%) lack this plateau and display activities throughout the entire CL, with ~2 (1 or 2) GAH-valleys per tachycardia. However, they did not characterize the SKYLINE pattern of reentry AT. In the present study, the aAFLs in post-AF ablation or post-surgery SHD displayed 1.5 (1 or 2) steep GAH-valleys.

In the study by Takigawa et al. [18], they demonstrated that deeper GAH-Valleys (usually ≦0.2) correlate with practical ablation sites well and should be investigated first. However, they also found practical ablation sites do not always correspond to regions highlighted by the deepest GAH-valley. It means that investigated regions by GAH ≤ 0.2 only could not identify all of the potential practical ablation site. From their study results on GAH-score versus practical ablation site, practical ablation site in GAH-score ≤ 0.20 and >0.2 was 62.8% and 15%, in patients with reentry AT excluding focal AT. With the threshold of GAH ≤ 0.2, practical ablation site was significantly correlated with GAH ≤ 0.2, comparing with GAH > 0.2. Hence, they concluded GAH-valley ≤ 0.2 should be investigated first. However, we could found that practical ablation site in GAH-valley ≤ 0.20 was only 62.8% in reentry AT. That is, investigated regions by GAH ≤ 0.20 was not able to find all of the potential practical ablation site. 

Our study had similar finding that successful ablation site was not always correspond to regions highlighted by GAH-valley ≤ 0.20. Successful site in GAH-score ≤ 0.20 and GAH 0.2–0.4 was 57% and 43% in patients with macro-reentry aAFL, excluding localized reentry. To identify all potential practical ablation site, we investigated GAH ≤ 0.20 first, followed by GAH 0.2–0.4.

### 4.4. LUMIPOINT Algorithm as an Automatic Search for Practical Ablation Sites

In the Lumipoint^TM^ algorithm, the surface of activated areas/time was strongly associated with electrical properties of the myocardium. The algorithm identified regions wherever the EGM timing fell within a narrow geographical area. GAH-Valleys are largely due to conduction ‘deceleration’ of the atrium participating in the reentry circuit.

Although wave-front collisions become more frequent with the conduction ‘deceleration’, the incidence of a slow conduction within the circuit remains high. About 87% of slow conducting areas are highlighted by GAH-Valley < 0.4 within the aAFL circuit. In addition, all of the successful termination sites correlated well with GAH-Valley < 0.4 (60%: GAH-scores < 0.2, 40%: GAH-scores 0.2 to 0.4). Such a good correlations allow clinicians quickly identify and further evaluate the potential ablation targets with a high sensitivity. This avoids missing any critical site with slow conduction and expedites mapping and the surgical procedures.

### 4.5. Clinical Implications

Our major finding is that the aAFL in post-AF ablation or prior surgery in SHD displays multiple deflected SKYLINE patterns and steeper GAH-Valleys. GAH-Valley represents abnormal conduction ‘deceleration’ in the reentry circuit. In the aAFL, such GAH-Valley scores (60%: GAH-scores < 0.2, 100%: GAH-scores < 0.4) were higher than those in the focal AT. In light of this insight from our findings, we proposed a flowchart to identify the potential ablation targets in patients with a complex aAFL in prior AF ablation or cardiac surgery in SHD (Figure 5).

### 4.6. Study Limitations

The main limitation of our study is the limited number of aAFL and patients. Another limitation is that our study was retrospectively performed without randomization and control group. We did not prospectively use the GAH/LUMIPOINT results to confirm its efficacy. Third, we did not use entrainment maneuver to confirm the mechanism of aAFL. The definition of the aAFL circuit and eletrophysiological phenomena was simply based on visualization of the activation, isochronal map and local EGM characteristics. Further clinical evaluation is necessary to validate these findings. Thus, a prospective randomized study is warranted to confirm whether or not it can facilitate and shorten aAFL ablation procedure time and improve outcomes.

Nevertheless, the purpose of the present study was to evaluate the feasibility and characterize Lumipoint^TM^ feature of complex aAFL with HDM Rhythmia^TM^ system.

## 5. Conclusions

Our present study shows the Lumipoint^TM^ algorithm efficiently detects slow conductions in aAFLs, identifies ablation sites, and terminates the aAFL with favorable outcomes.

## Figures and Tables

**Figure 1 jpm-12-01102-f001:**
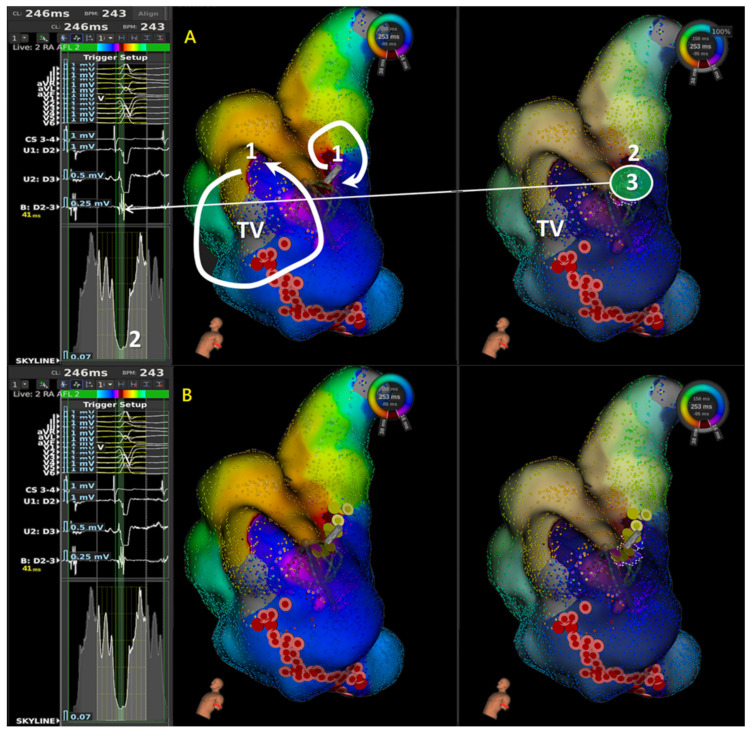
Illustration of stepwise approach in application of LUMIPOINT & Skyline for aAFL in a patient with prior cardiac surgery. In frame (**A**), 1 denotes first step to determine the possible main circuit in isochronal map; 2 denotes second step to scan the Skyline, identify the GAH-valley, and check the corresponding highlighted area in isochronal map; and 3 denotes third step to confirm the wave-front property and characteristics of local signals in highlighted area. In frame (**B**), ablation lesion set showed initial cavo-tricuspid isthmus (CTI) linear ablation failed to terminate the atrial flutter; instead, focal ablation at the highlighted area by Skyline valley successfully terminated atrial flutter to sinus rhythm (SR). In the first step, isochronal map showed two reentry circuits, one with macro-reentry counterclockwise atrial flutter and the other with localized reentry confined at mid-septum. Without entrainment, we cannot confirm the mechanism. Therefore, we started with CTI linear ablation instead of focal ablation because macro-reentry CTI dependent atrial flutter was more prevalent, and focal ablation in mid-septum took more risk in AV node injury. Since activation sequence and tachycardia cycle length did not change after CTI linear ablation, focal ablation in mid-septum terminated atrial flutter to SR. In this case, limitation of activation and isochronal map were seen and Skyline in LUMIPOINT feature adds its value and helped identify the slow conduction. (TV: Tricuspid Valve).

**Figure 2 jpm-12-01102-f002:**
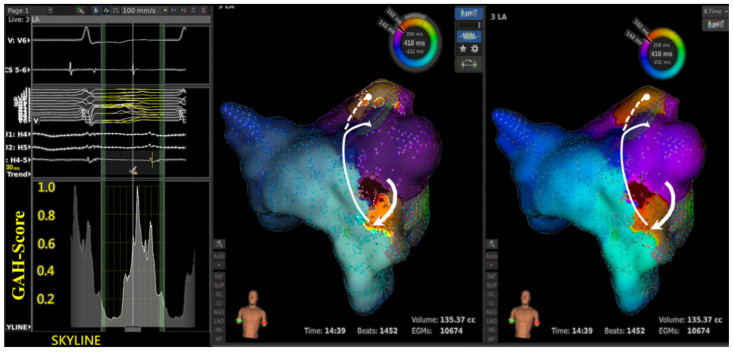
A case of localized reentry aAFL.

**Figure 3 jpm-12-01102-f003:**
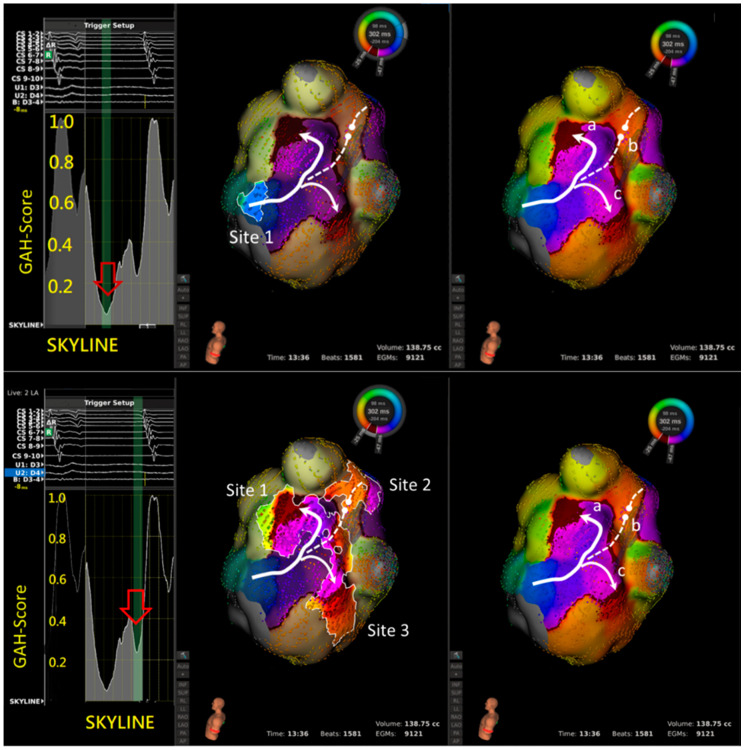
A case of multiple-loop (figure-of-eight) macro-reentry aAFL. Right panel shows wave-front propagating through a slow conduction region at the lateral mitral annulus before diverging into three different activation wave-fronts; critical slow conduction along with LSPV ridge (**a**), wave-front collision (**b**), and another region of slow conduction along the LIPV ridge (**c**).

**Figure 4 jpm-12-01102-f004:**
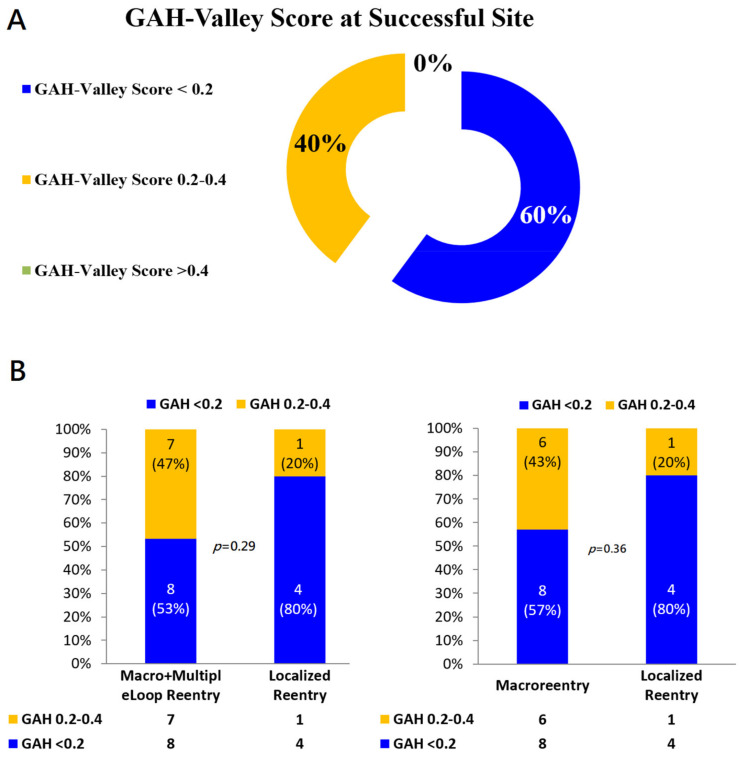
(**A**) GAH-Valley Score at the Successful Site. (**B**) Distributions of GAH-Scores in different types of Reentry aAFLs. In the localized reentry, the highlighted area of GAH-Valley scores < 0.2 significantly matched with the successful ablation sites, compared with both macro- and multiple-loop reentry aAFLs (Left). Likewise if excluding the multiple-loop reentry (Right).

**Figure 5 jpm-12-01102-f005:**
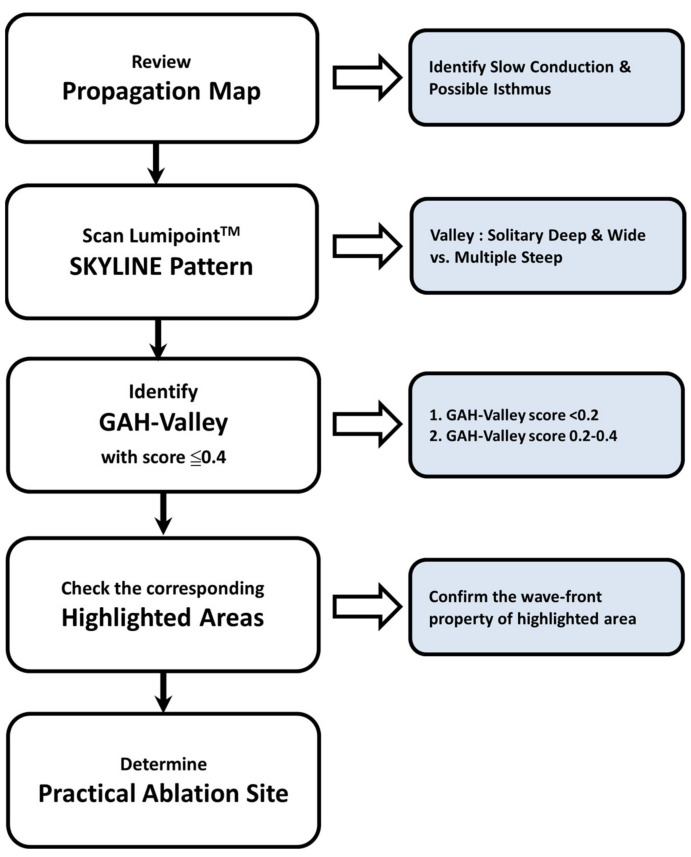
The proposed stepwise approach in Lumipoint^TM^ module to identify practical ablation sites in patients with aAFL.

**Table 1 jpm-12-01102-t001:** Patients’ Clinical Characteristics (*N* = 15) and Atypical Atrial Flutter Characteristics.

Clinical Charateristics	
Age (years)	65.3 ± 10.7
Male (*n*,%)	9 (60%)
SHD	9 (60%)
CHF	6 (40%)
Hypertension	6 (40%)
Diabetes Mellitus	3 (20%)
CAD	5 (33%)
Stroke	1 (7%)
CHA2DS2-VASc score	2.6 ± 2
LVEF (%)	52.1 ± 10.8%
Prior Procedure	12 (80%)
Post-cardiovascular Surgery	7 (47%)
Post-AF/AFL Ablation	6 (40%)
No. of Previous Procedure	0.8 ± 1.0
PVI	6 (40%)
Roof Line	5 (33%)
Mitral Isthmus Line	5 (33%)
Cavo-tricuspid Isthmus Line	8 (53%)
Defragmentation in LA (CFAE ablation)	1 (7%)
Follow-up (m)	12.5 ± 9.3
aAFL characteristics (*n* = 20)	*n*(%)
Macro-reentry AFL	14 (70%)
Anatomical AFL	8 (40%)
Peri-mitral	4
Peri-tricuspid	1
Roof-dependent	2
Between Aortic groove & mitral	1
Surgical Scar-related	3 (15%)
Other atypical AFL (gap related, inter-atrial septum)	3 (15%)
Localized reentry (micro-reentry)	5 (25%)
Multiple loop AFL	1 (5%)

aAFL: atypical atrial Fluter; AFL: atrial flutter; SHD: Structure heart disease; CHF: congestive heart failure, CAD: coronary artery disease, CHA2DS2-VASc = congestive heart failure, hypertension, age 75 years or older, diabetes mellitus, previous stroke or transient ischemic attack, vascular disease, age 65 to 74 years, female; LVEF: left ventricular ejection fraction; AF: atrial fibrillation; AFL: atrial flutter, PVI: pulmonary vein isolation; CFAE: complex fractionated atrial electrograms.

**Table 2 jpm-12-01102-t002:** Highlighted areas and lowest GAH-Valley in LUMIPOINT algorithm.

	Total No. of GAH-Valley	No. of GAH-Valley per AFL	No. of Highlighted Areas	No. of Highlighted Areas per GAH-Valley	No. of Highlighted Areas per AFL	Successful Termination Site Corresponding to Highlighted Area
Macroreentry AFL (*n* = 14)	27	2 (1-2)	51	2 (1-2)	3 (2–5)	8/14
Localized reentry AFL (*n* = 5)	6	1 (1–1)	11	1.5 (1–3)	1 (1–3)	4/5
Multiple loop AFL (*n* = 1)	1	1	1	1	1	0/1
Total (*n* = 20)	34	1.5 (1-2)	63	2 (1-2)	3 (1–4.5)	12/20

Values are given as *n*, median (25th percentile-75th percentile), or *n*/*n* (%).

**Table 3 jpm-12-01102-t003:** Electrophysiological properties of highlighted areas based on the lowest GAH-Valley.

(A) 14 Macroreentry AFL	Slow Conduction (*n* = 23)	Wavefront Collision (*n* = 20)	Line of Block (*n* = 5)	Others (*n* = 3)
In the circuit (*n* = 20)	19/23 (83)	0	0	1/3 (33)
Out of the circuit (*n* = 31)	4/23 (17)	20/20 (100)	5/5 (100)	2/3 (67)
Total (*n* = 51)	23 (45)	20 (39)	5 (10)	3 (6)
(B) 5 Localized AFL	Slow Conduction (*n* = 6)	Wavefront Collision (*n* = 5)	Line of Block (*n* = 0)	Others (*n* = 0)
In the circuit (*n* = 6)	6/6 (100)	0/5 (0)	0	0
Out of the circuit (*n* = 5)	0/6 (0)	5/5 (100)	0	0
Total (*n* = 11)	6 (55)	5 (45)	0	0
(C) 1 Multiple loop AFL	Slow conduction (*n* = 1)	Wavefront Collision (*n* = 0)	Line of Block (*n* = 0)	Others (*n* = 0)
In the circuit (*n* = 1)	1/1 (100)	0	0	0
Out of the circuit (*n* = 1)	0/0 (0)	0	0	0
Total (*n* = 2)	1/1 (100)	0	0	0
(D) Total 20 AFL	Slow conduction (*n* = 30)	Wavefront Collision (*n* = 26)	Line of Block (*n* = 5)	Others (*n* = 3)
In the circuit (*n* = 27)	26/30 (87)	0/26 (0)	0/5 (0)	1/3 (33)
Out of the circuit (*n* = 37)	4/30 (13)	26/26 (100)	5/5 (100)	2/3 (67)
Total (*n* = 64)	30 (47)	26 (41)	5 (8)	3 (5)

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
