# Peer review of "Patterns and Characteristics of SKYLINE-Lumipoint Feature in the Catheter Ablation of Atypical Atrial Flutter: Insight from a Novel Lumipoint Module of Rhythmia Mapping System"

_jpm, 2022, doi:10.3390/jpm12071102_

Round 1
Reviewer 1 Report
Dear Sir/Madam,
I had the opportunity to act as a reviewer on the recent submission by Li et al. to the Journal of Personalized Medicine.
The authors present interesting research evaluating the feasibility and characterizing the Lumipoint feature of complex atypical atrial flutter using high density mapping (Rhythmia system). They have included a total of 15 patients undergoing catheter ablation of atypical atrial flutter. The authors found that successful sites of ablation matched highlighted areas based on global activation histogram (GAH) valley scores <0.4. The manuscript is well written and the results are interesting and of some clinical interest.
However, following issues need to be addressed:
- The central issue of the manuscript is its novelty. As the authors state (lines 328-333), the same feature (Skyline) was studied in the prospective study by Takigawa et al. (Heart Rhythm 2019), which included 67 patients. Of these 67 patients 57 patients had complex postprocedural (pulmonary vein isolation and cardiovascular surgery) and of the 108 mapped atrial tachycardias only 17 were focal in nature, whereas 79 were complex macro-reentry, localized reentry and multiple loop (12 CTI dependent excluded). Takigawa et al. concluded that GAH valleys ≤0.2 correlate with practical ablation sites and should be investigated first, highlighting it also in their Figure 4B (after excluding focal atrial tachycardias).
Therefore, the element of novelty of this manuscript is that also GAH valley scores 0.2-0.4 should be investigated.
Could the authors please expand their comments on this issue?
Minor issues:
1- Line 37 (abstract): instead of “focal” is meant “localized reentry”?
2- Line 56: please explain or reformulate “extend beyond the common understanding and simple categorization”.
3- Line 63: “atrial procedures [17,18] and none so far on complex aAFLs”. The majority of the patients in the study by Takigawa et al. included complex aAFL. Can you please comment on that?
4- Line 70: “These were…” seems like a poor formulation, could the authors rephrase?
5- Line 84: the authors mean left atrial thrombus exclusion? This is not always appendage.
6- Line 134: what exactly do the authors mean by “suspected activity”?
7- Figure 1 is referred to in the Methods section, but it also includes results. Could the authors clearly separate methods from results?
8- Did the authors perform a map in sinus rhythm after treating the tachycardia: “These were consistent with the complexity of underlying anatomy or substrates” (lines 283-284)?
9- The authors mention on line 301 the “conduction deceleration”. How could that be translated into a quantitatively decrease in conduction velocity?
Best regards,
Author Response
Thank you very much for your detailed and instructive comments, and we appreciate your comments and for allowing us to revise this manuscript. The responses to those comments are dictated below.
Response to the reviewer 1:
Thank you very much for your detailed and instructive comments, and we appreciate your comments and for allowing us to revise this manuscript. The responses to those comments are dictated below.
- The central issue of the manuscript is its novelty. As the authors state (lines 328-333), the same feature (Skyline) was studied in the prospective study by Takigawa et al. (Heart Rhythm 2019), which included 67 patients. Of these 67 patients 57 patients had complex postprocedural (pulmonary vein isolation and cardiovascular surgery) and of the 108 mapped atrial tachycardias only 17 were focal in nature, whereas 79 were complex macro-reentry, localized reentry and multiple loop (12 CTI dependent excluded). Takigawa et al. concluded that GAH valleys ≤0.2 correlate with practical ablation sites and should be investigated first, highlighting it also in their Figure 4B (after excluding focal atrial tachycardias).
Therefore, the element of novelty of this manuscript is that also GAH valley scores 0.2-0.4 should be investigated.
Could the authors please expand their comments on this issue?
Reply: We appreciate your perspective and comments on this important issue. And also thanks for allowing us to expand our finding on this issue.
In the study by Takigawa et al. (Heart Rhythm 2019), they demonstrated that deeper GAH-Valleys (usually ≦0.2) correlate with practical ablation sites well and should be investigated first. However, they also found practical ablation sites do not always correspond to regions highlighted by the deepest GAH-valley. It means that investigated regions by GAH ≦0.2 only could not identify all practical ablation site.
From their study results on GAH-score versus practical ablation site, practical ablation site in GAH-score ≦0.20 and >0.2 was 62.8% and 15%, in patients with reentry AT, excluding focal AT. With the threshold of GAH ≦0.2, practical ablation site was significantly correlated with GAH ≦0.2, comparing with GAH >0.2. Hence, they concluded GAH-valley ≦0.2 should be investigated first. However, we found that practical ablation site in GAH-valley ≦0.20 was only 62.8% in reentry AT. That is, investigated regions by GAH≦0.20 was not able to find all practical ablation site.
Our study had similar finding that successful ablation site was not always correspond to regions highlighted by GAH-valley ≦0.20. Successful site in GAH-score ≦0.20 and GAH 0.2-0.4 was 57% and 43% in patients with macro-reentry aAFL, excluding localized reentry. To identify all practical ablation sites, we investigated GAH≦0.20 first, followed by GAH 0.2-0.4.
We therefore add the text on Page 11, line 334-350, as ‘‘In the study by Takigawa et al.[18], they demonstrated that deeper GAH-Valleys (usually ≦0.2) correlate with practical ablation sites well and should be investigated first. However, they also found practical ablation sites do not always correspond to regions highlighted by the deepest GAH-valley. It means that investigated regions by GAH ≦0.2 only could not identify all of the potential practical ablation site. From their study results on GAH-score versus practical ablation site, practical ablation site in GAH-score ≦0.20 and >0.2 was 62.8% and 15%, in patients with reentry AT excluding focal AT. With the threshold of GAH ≦0.2, practical ablation site was significantly correlated with GAH ≦0.2, comparing with GAH >0.2. Hence, they concluded GAH-valley ≦0.2 should be investigated first. However, we found that practical ablation site in GAH-valley ≦0.20 was only 62.8% in reentry AT. That is, investigated regions by GAH≦0.20 was not able to find all of the potential practical ablation site.
Our study had similar finding that successful ablation site was not always correspond to regions highlighted by GAH-valley ≦0.20. Successful site in GAH-score ≦0.20 and GAH 0.2-0.4 was 57% and 43% in patients with macro-reentry aAFL, excluding localized reentry. To identify all potential practical ablation site, we investigated GAH≦0.20 first, followed by GAH 0.2-0.4.’’
Minor issues:
- Line 37 (abstract): instead of “focal” is meant “localized reentry”?
Reply: Thanks for your reminding and we’re sorry for this error. We corrected the ‘‘focal’’ to ‘‘localized-reentry’’ aAFL. (on Page 1, Line 37)
- Line 56: please explain or reformulate “extend beyond the common understanding and simple categorization”.
Reply: Thanks for your comment and suggestion and we are sorry for our unclear presentations. We reformulate this sentence and make it easier to understand. We revised this sentence on Page 2, line 55-58, as ‘‘In aAFL ablation, high-density mapping (HDM) system reveals complex activation patterns and extends the common knowledge of AFL. It provides more information about tissue and structure than simple categorization of them as focal, localized reentry or macro-reentry [11,12].’’
- Line 63: “atrial procedures [17,18] and none so far on complex aAFLs”. The majority of the patients in the study by Takigawa et al. included complex aAFL. Can you please comment on that?
Reply: We’re sorry for the incorrect expression and thanks for your comments. We rephrased ‘‘…and none so far on complex aAFLs’’ to ‘‘not to speak of complex aAFLs’’ on Page 2, Line 64.
- Line 70: “These were…” seems like a poor formulation, could the authors rephrase?
Reply: Thanks for your correction and sorry for this typo. We corrected the phrase ‘‘These were’’ to ‘‘There were’’ on Page 2, Line 71.
- Line 84: the authors mean left atrial thrombus exclusion? This is not always appendage.
Reply: Thanks for your reminding and correction. We corrected the ‘‘left atrial appendage thrombus’’ to ‘‘left atrial thrombus’’ on Page 2, Line 84.
- Line 134: what exactly do the authors mean by “suspected activity”?
Reply: Thanks for your correction. We revised the sentence ‘highlighting regions of the map showing the ‘‘suspected activity’’ in a given time-of-interest period’’ and rephrased as ‘highlighting regions of the map containing electrogram activity in a given time-of-interest period ’ on Page 3, Line 135.
- Figure 1 is referred to in the Methods section, but it also includes results. Could the authors clearly separate methods from results?
Reply: Thanks for your suggestion and correction. We’re sorry for this erroneous arrangement. To separate methods from results and make the text easy to understand, we changed original Figure 2 to Figure 1 and put it in the Methods section, and changed original Figure 1 to Figure 2 and put it in the Results section.
- Did the authors perform a map in sinus rhythm after treating the tachycardia: “These were consistent with the complexity of underlying anatomy or substrates” (lines 283-284)?
Reply: Thanks for your comments and suggestions. We did not routinely perform a post-ablation sinus map after treating the aAFL. Therefore, we agreed that the sentence ‘‘These were consistent with ….’’ speaks in a strong term. We rephrased the sentence ‘‘These were consistent with the complexity…..’’ to ‘‘This suggests that complexity of underlying anatomy or substrates caused complex aAFL circuits’’ on Page 2, Line 71.
- The authors mention on line 301 the “conduction deceleration”. How could that be translated into a quantitatively decrease in conduction velocity?
Reply: Thanks for your comments and suggestions. We thought the ‘‘conduction deceleration’’ could not be translated into a quantitatively decrease in conduction velocity. The possible reason might be as follows: In Skyline feature, the steep drop in GAH-score means the quantitatively decrease in global atrial depolarization region, not necessarily to be the conduction velocity. In a simple circuit, the drop in GAH-score means a quantitatively decrease in depolarization area, that could translate into a decrease in conduction velocity. In a complex circuit, however, the decrease in global atrial depolarization region could result from a decrease in conduction velocity of the main circuit, a decrease in conduction termination of a bystander circuit (ie. line of block or wave-front collision area), or both. For this reason, we thought a conduction deceleration could not be simply translated into a quantitatively decrease in conduction velocity.
Reviewer 2 Report
Li et al performed a novel Lumipoint algorithm in the Rhythmia mapping system to evaluate tachycardia circuit. The paper is well written and is very interesting. The Authors should be congratulated for their effort in assessing the mechanisms of atypical atrial flutter. The methods are well described and data presentation is clear. However, there are some criticisms which need to be clarified.
1- Patients who experienced atrial flutter recurrences should be described in the text. The authors should describe the type of arrhythmia and the strategy adopted for the its management.
2- In Figure 2 the authors should report the meaning of "TV".
Author Response
Thank you very much for your detailed and instructive comments, and we appreciate your comments and for allowing us to revise this manuscript. The responses to those comments are dictated below.
Response to the reviewer 2:
Thank you very much for your detailed and instructive comments, and we appreciate your comments and for allowing us to revise this manuscript. The responses to those comments are dictated below.
- Patients who experienced atrial flutter recurrences should be described in the text. The authors should describe the type of arrhythmia and the strategy adopted for the its management.
Reply: Thanks for your comments and suggestions. About the patients who experienced atrial flutter recurrences, 5 of 15 patients (33%) developed atrial flutter/fibrillation recurrence at a mean of 12.5 months follow-up, including 4 atypical atrial flutters and one atrial fibrillation. On Page 5, Line 198-199, we described it as ‘‘At a mean follow-up period of 12.5 ±9.3 months, 5 of 15 patients (33%) developed AF/AFL recurrences (4 aAFLs, 1 AF).’’ We marked it up in yellow color on Page 5, Line 198-199..
Our strategy for aAFL recurrence was usually radiofrequency catheter ablation and for Af recurrence was rhythm control if symptomatic. In these 4 recurrent aAFL and one recurrent Af cases, patients refused further ablation and received anti-arrhythmic drugs during follow-ups. Since these 5 patients with recurrences did not receive further ablation, we are not able to provide more details because the pharmacological treatment after recurrence was not the issue we discussed in this article.
- In Figure 2 the authors should report the meaning of "TV".
Reply: Thanks very much for your correction and we’re sorry for the missing information for TV in Figure 2. We add ‘‘TV: Tricuspid Valve’’ in the Figure 2 Legend. Because we switched the order of Figures 1 and 2 in the revised manuscript, the correlated change [TV: Tricuspid Valve] is in the Figure 1 in the revised manuscript. (Page 5, Line 177)